## 1 Multi-sensor in situ observations to resolve the sub-mesoscale features in

# 2 the stratified Gulf of Finland, Baltic Sea

U. Lips, V. Kikas, T. Liblik, and I. Lips

Marine Systems Institute at Tallinn University of Technology, Akadeemia Road 15a, 12618

- Tallinn, Estonia
- Correspondence to: U. Lips (urmas.lips@msi.ttu.ee)

## 9 Abstract

High-resolution numerical modeling, remote sensing and in situ data have revealed significant 10 role of sub-mesoscale features in shaping the distribution pattern of tracers in the ocean upper 11 layer. However, in situ measurements are difficult to conduct with the required resolution and 12 coverage in time and space to resolve the sub-mesoscale, especially in such relatively shallow 13 basins as the Gulf of Finland where the typical baroclinic Rossby radius is 2-5 km. To map 14 the multi-scale spatiotemporal variability in the gulf, we initiated continuous measurements 15 with autonomous devices, including a moored profiler and Ferrybox system, which were 16 complemented by dedicated research vessel based surveys. The analysis of collected high-17 resolution data in summers 2009-2012 revealed pronounced variability at the sub-mesoscale 18 19 in the presence of mesoscale upwelling/downwelling, fronts, and eddies. The horizontal wavenumber spectra of temperature variance in the surface layer had slopes close to -2 20 21 between the lateral scales from 10 to 0.5 km. Similar tendency towards the -2 slopes of horizontal wavenumber spectra of temperature variance was found in the seasonal 22 23 thermocline between the lateral scales from 10 to 1 km. It suggests that the ageostrophic submesoscale processes could contribute considerably to the energy cascade in such stratified sea 24 25 basin. We showed that the intrusions of waters with different salinity, which indicate the occurrence of layered flow structure, could appear in the process of upwelling/downwelling 26 development and relaxation in response to variable wind forcing. We suggest that the sub-27 mesoscale processes play a major role in feeding surface blooms in the conditions of coupled 28 29 coastal upwelling and downwelling events in the Gulf of Finland. 30

Keywords: sub-mesoscale features, stratification, autonomous systems, spatial spectra, Gulf
of Finland

33

### 34 1. INTRODUCTION

35

Essential contribution of mesoscale processes to the vertical exchanges of nutrients in the 36 open ocean has been suggested and proved by a number of studies in the recent two decades 37 (e.g. McGillicuddy et al., 1998; Martin and Pondaven, 2003). These studies were motivated 38 by the discrepancies between the direct measurements of vertical turbulent exchanges and 39 indirect estimates of nutrient fluxes to support net primary production (Jenkins, 1988). Two 40 conceptual views of additional nutrient supplies related to mesoscale eddies exist: (1) vertical 41 42 exchanges due to the time evolution of eddies and (2) vertical pumping at small scales, i.e. within the sub-mesoscale structures (Klein and Lapeyre, 2009). The latter hypothesis is 43 44 supported by recent observations and modeling with increased spatial resolution suggesting that the sub-mesoscale processes significantly contribute to the vertical exchange of water 45 46 mass properties between the upper and deep ocean (Bouffard et al., 2012). Sub-mesoscale processes are characterized by order-one (O(1)) Rossby and Richardson numbers (Thomas, 47 48 2008), large vertical velocity and vorticity fluctuations and large vertical buoyancy flux, resulting in considerable intermittency of oceanographic properties in the upper ocean (Capet 49 50 et al., 2008).

Main physical forcing components for the non-tidal Baltic Sea system are the atmospheric 51 forcing, exchange of heat energy and fresh water through the sea surface, and input of 52 freshwater from rivers and saltier North Sea water through the Danish Straits (Omstedt et al., 53 2004). It was identified already in the 1980s that the Baltic Sea has rich mesoscale variability 54 with spatial scales O(10) km through the whole water column (Aitsam et al., 1984) and 55 56 evidence is increasing that remarkable changes occur in the system due to meso- and sub-57 mesoscale processes (e.g. Nausch et al., 2009; Lips et al., 2009). Recent results based on analysis of high resolution in situ (Lips et al., 2011), numerical modeling (Laanemets et al., 58 59 2011) and remote sensing (Uiboupin et al., 2012) data from the Gulf of Finland showed that the sub-mesoscale features significantly shape the distribution pattern of tracers in this 60 61 stratified basin. Among such features, the upwelling filaments and intra-thermocline intrusions with lateral scales less than the internal Rossby radius of deformation, which is 62 63 about 2-5 km in the Gulf of Finland (Alenius et al., 2003), are named. The layered structure of the major basins of the Baltic Sea, with the seasonal thermocline 64

and the halocline situated at different depths – about 10-30 m and 60-80 m, respectively, is a
challenge to be accurately described by numerical models (Tuomi et al., 2012). In many
cases, a proper validation of model results is difficult due to the absence of observational data

with the required resolution and coverage in time and space. In order to fill this gap a number 68 of autonomous devices, including moored profilers and Ferryboxes, and towed instruments 69 are applied in the Gulf of Finland. According to high-resolution profiling at a fixed position in 70 71 the Gulf of Finland, quasi-stationary stratification patterns of the thermocline occurred there 72 at time scales of 4-15 days (Liblik and Lips, 2012) and the vertical dynamics of phytoplankton were largely defined by these patterns (Lips et al., 2011). Furthermore, TS-73 74 variability at the sub-mesoscale was significant during the transition periods between the 75 quasi-stationary patterns (Liblik and Lips, 2012).

Coastal upwelling events are prominent mesoscale features in the Gulf of Finland 77 (Uiboupin and Laanemets, 2009) leading to considerable vertical transport of nutrients into 78 the euphotic layer (Laanemets et al., 2011; Lips et al., 2009) and influencing the phytoplankton growth and species composition (e.g. Lips and Lips, 2010). Analysis of 79 80 Ferrybox data collected along the ferry line Tallinn-Helsinki in the central part of the Gulf of Finland revealed occurrence of the two types of upwelling events (Kikas and Lips, 2015). 81 82 Beside of the classical coastal upwelling with a strong upwelling front, the second type of upwelling events existed where a gradual decrease of surface layer temperature from the open 83 84 sea towards the coast was observed. The latter type was characterized by a relatively high spatial variability at scales of a few to ten kilometers, which as suggested by Kikas and Lips 85 (2015) could be a sign of sub-mesoscale dynamics in the case of wind forcing not strong 86 enough to produce an Ekman transport in the entire surface layer. This suggestion of higher 87 sub-mesoscale activity associated with some types or phases of coastal upwelling has to be 88 89 analyzed further. Such analysis based on combined Ferrybox, buoy profiler and Scanfish data was one of the tasks of the present study. 90

According to the theory of quasi-geostrophic turbulence, the shape of the energy spectrum 91 92 should follow the -3 slope in the logarithmic scale at the spatial scales below the mesoscale 93 (Charney, 1971). It has been shown that if the spatial resolution of numerical models was increased the spectral slope converted rather to -2 than -3 (Capet et al., 2008) suggesting that 94 95 sub-mesoscale processes play an important role in the energy cascade from larger to smaller scales. Still, it is a major challenge to map sub-mesoscale processes and phenomena by in situ 96 97 observations. Due to the temporal and spatial scales to be resolved, distinction between the temporal and spatial variability is difficult based on the high-resolution 3-D surveys by a 98 single technique, platform or device. We have applied in situ observations, using both 99 autonomous devices and research vessel, for mapping temporal variability in temperature, 100 101 salinity and chlorophyll a distribution patterns in the Gulf of Finland. Close to the Ferrybox

line Tallinn-Helsinki, an autonomous profiler was deployed in the summers of 2009-2012.
This dataset allows us to estimate the temporal changes in the horizontal distribution patterns
in the surface layer and vertical stratification (vertical temperature and salinity distribution) at
a station close to the Ferrybox line simultaneously. In addition, Scanfish surveys were
conducted in the area to reveal the spatial variability in the sub-surface layer.

The main aim of the present paper is to describe spatial and temporal variability at the mesoscale and sub-mesoscale, indicate the main sub-mesoscale features and their effects on 108 the vertical stratification as well as chlorophyll a dynamics under different forcing conditions 109 110 by combining high-resolution observational data (Ferrybox, buoy profiler, and Scanfish). We would like to demonstrate that multi-sensor in situ observations, initiated to meet the data 111 112 needs in operational oceanography, are able to resolve the sub-mesoscale features and are a good basis for descriptive and statistical analysis of mesoscale and sub-mesoscale 113 114 variability/features in the Gulf of Finland. The hypothesis that under certain mesoscale conditions, such as development and relaxation of coastal upwelling events in a stratified 115 116 estuary, the sub-mesoscale processes are more energetic than predicted by the theory of quasigeostrophic turbulence in the ocean interior is tested. 117

#### 119 2. MATERIAL AND METHODS

#### 120 **2.1 Measurement systems and data**

The dataset analyzed in the present study was gathered using an observational network applied by the Marine Systems Institute at Tallinn University of Technology in the Gulf of Finland, Baltic Sea. It includes autonomous measurements and sampling on board a ferry traveling between Tallinn and Helsinki and autonomous measurements at a profiling buoy station close to the ferry route. Additionally, research vessel based measurements and sampling, as well as surveys using a towed undulating vehicle (Scanfish), are employed (Fig. 1).

The Ferrybox system records temperature (T), salinity (S), and chlorophyll *a* (Chl *a*) 128 129 fluorescence in the surface layer (water intake is approximately at 4 m depth) twice a day along the ferry route Tallinn-Helsinki (the system is described in detail by Kikas and Lips, 130 131 2015). The time resolution of measurements of 20 s corresponds to an average spatial 132 resolution of 160 m. For temperature measurements, a PT100 temperature sensor with a measuring range from -2 to +40 °C and accuracy of  $\pm 0.1\%$  of the range is used. The sensor is 133 installed close to the water intake to diminish the effect of warming of water while flowing 134 135 through the tubes onboard. For salinity measurements a FSI Excell thermosalinograph

- (temperature and conductivity meter) is used, and the data quality is checked by water
- sampling and analysis of samples by a high-precision salinometer Portasal 8410A (Guildline
- Instruments) 2-4 times a year. For Chl *a* fluorescence and turbidity (turbidity data not
- presented here) measurements, a SCUFA submersible fluorimeter (Turner Designs) with a
- flow-through cap is used. Acid-washing cleaning system is applied to prevent biofouling and
- 11-17 water samples along the ferry route are collected once a week for laboratory analysis of
- Chl *a* content to calibrate the fluorimeter data.

Figure 1. Map of the Baltic Sea (left panel) and the study area (right panel). Black lines indicate the Ferrybox
route between Tallinn and Helsinki, blue line the Scanfish track on 22 July 2010, 2 August 2010, 4 July 2012 and
20 July 2012, green line the Scanfish track on 27 July 2010 and red line the Scanfish track on 31 July 2012.
Yellow dots indicate the location of the buoy station AP5 and the Kalbadagrund meteorological station.

The autonomous profiler deployed in the summers of 2009-2012 at station AP5 (Fig. 1) recorded vertical profiles of temperature, salinity, and Chl a fluorescence in the water layer 151 from 2 to 50 m with a time resolution of 3 h and a vertical resolution of 10 cm. The sensor set 152 153 at the buoy profiler consisted of an OS316plus CTD probe (Idronaut S.r.l.) equipped with a Seapoint Chl a fluorimeter. To avoid biofouling of sensors, the parking depth well below the 154 euphotic layer depth and electrochemical antifouling system were applied. Ship-borne 155 measurements and sampling close to the buoy profiler were arranged bi-weekly to check the 156 quality of data (compare the vertical profiles from the buoy with those from the research 157 vessel) and to calibrate the Chl a fluorimeter by laboratory analyses of Chl a content from the 158 159 water samples.

The dataset used also includes Scanfish surveys of temperature, salinity, and Chl a 160 161 fluorescence conducted to map the horizontal distribution of T, S, and Chl a in the water column from 2 to 45 m (see location of sections in Fig. 1). The average distance between the 162 consecutive Scanfish cycles, including down- and upcast while the vessel was moving with a 163 speed of 7 knots, was 600 m. Data was recorded continuously (both down- and upcast are 164 used) and the processed data were stored with a vertical resolution of 0.5 m. Scanfish sensor 165 set consisted of a Neil Brown Mark III CTD probe and TriOS microFlu-chl-A fluorimeter. 166 167 Ship-borne CTD measurements and water sampling was conducted before and after the 168 Scanfish surveys to control the quality of Scanfish data and calibrate the fluorimeter. 169 To calibrate the used (different) Chl a sensors, the Chl a concentration in the water 170 samples was determined in the laboratory. Whatman GF/F glass fiber filters and extraction at room temperature in the dark with 96% ethanol for 24 h were used. The Chl a content from 171 172 the extract was measured spectrophotometrically (HELCOM, 1988) by Thermo Helios  $\gamma$ . The dataset from July-August 2009-2012 analyzed in the present study is described in 173 174 Table 1. Altogether data from 461 ferry crossings Tallinn-Helsinki, 968 CTD and Chl a

175 profiles collected at station AP5 and six Scanfish surveys are included.

### 177 **2.2 Calculations**

The results in the following sections are presented as graphs of pre-processed observational 178 data and horizontal wavenumber spectra of temperature variance calculated from the Ferrybox 179 and Scanfish measurements as well as the estimated characteristics of vertical stratification at 180 station AP5. The use of spatial spectra of temperature (instead of density) was based on the 181 assumptions that in summer in the surface and thermocline layer of the GoF the water density 182 is mainly controlled by temperature and it is measured by one sensor while density has to be 183 estimated from the readings of two separate sensors. The following approaches are used in the 184 calculations. 185

Horizontal wavenumber spectra of temperature variance were calculated for each ferry 186 187 crossing between Tallinn and Helsinki assuming that the distance between the data points along the ferry route was constantly 160 m. The areas close to the harbors, where the ferry 188 189 speed was varying, were excluded, and only the data along the ferry route between the 190 latitudes 59.48 N and 60.12 N were used. The mean spectra for a certain period with quasi-191 stationary variability were obtained by averaging of single spectra over this period. The spectral slopes between the spatial scales of 10 and 0.5 km were estimated. The overall 192 193 variability was characterized by daily standard deviations of temperature along the ferry route.

Horizontal wavenumber spectra of temperature variance in the sub-surface layer were 194 calculated using the data of Scanfish surveys. Since the distance between the consecutive 195 profiles varies depending on the depth, the Scanfish data were first interpolated to the grid 196 with a constant horizontal step of 300 m, which corresponds to the average distance between 197 the up- and downward casts. Then the individual spectra for every depth (with 0.5 m step) 198 were calculated, and the mean spectra in 10 m thick water layers were obtained by averaging 199 all spectra in those layers containing 21 individual spectra. The spectral slopes between the 200 201 spatial scales of 10 and 1 km were estimated.

Vertical stratification was described by estimating the potential energy anomaly *P* (Simpson and Bowers, 1981; Simpson et al., 1990) as:

$$P = \frac{1}{h} \int_{-h}^{0} (\rho_A - \rho) gz dz, \quad \rho_A = \frac{1}{h} \int_{-h}^{0} \rho dz; \quad (1)$$

where  $\rho(z)$  is the density profile over the water column of depth *h*. The stratification parameter *P* (J m<sup>-3</sup>) is the work required to bring about the complete mixing of the water column under consideration. Similarly to Liblik and Lips (2012), the integration was conducted from the sea surface until 40 m depth. If the surface data were missing (upper two meters where the buoy profiler did not measure), the uppermost available density value was extrapolated to the surface.

Intrusion index was calculated as a sum of negative salinity gradients (g kg<sup>-1</sup> m<sup>-1</sup>) in the 210 211 water layer from the sea surface to 40 m depth. Before calculations, the salinity profiles were smoothed by 2.5 m window. The idea behind the method comes from the fact that on the 212 213 background of vertical salinity gradient with a fresher surface layer and more saline deep layer, lateral salinity gradients exist in the study area. In general, fresher waters originate from 214 215 the east (the Neva River and other larger rivers in the GoF) while more saline waters originate from the Baltic Proper. This general lateral salinity gradient could be enhanced locally as a 216 217 result of meso- and sub-mesoscale dynamics. If the water layers with a thickness of a few to 218 10 meters move in different directions, vertical salinity inversions could be generated in the water column where the vertical density gradient is mostly maintained by the temperature 219 distribution. Thus, high values of intrusion index indicate the occurrence of layered flow 220 221 structures.

The Chl *a* fluorescence data acquired with different sensors attached to the Ferrybox system, buoy profiler, and Scanfish were converted into Chl *a* content values using equations of linear regression between the fluorescence readings and results of laboratory analyses of water samples. The conversation equation of Chl a = 2.47 x F ( $r^2 = 0.41$ , p < 0.05) was used for the buoy profiler fluorescence data analyzed in this paper to convert fluorescence (F; in

- arbitrary units) into Chl a content in mg m<sup>-3</sup>. Interpretation of Ferrybox fluorescence data was
- sometimes difficult due to some problems with biofouling. In the present study, we used only
- data from July 2010 when the found regression line had the following parameters: Chl a =
- 2.34 x F 2.41 ( $r^2 = 0.77$ , p