# Peer review of "the stratified Gulf of Finland, Baltic Sea"

_Ocean Science, 2016_

## Referee Comment (RC1) · F. Colijn (Referee) · 2 Mar 2016

This MS is a very useful contribution to the understanding of physical processes in a coastal sea. The main issue is the use of different observational techniques like moorings, Surveys and Ferryboxes to obtain high Resolution high frequency data on physical and biological Parameters.The paper is well written and easily understandable. There are some language issues which Need to be solved by a native Speaker and one Point of criticism should be taken on board by the authors. The paper is very descriptive, thus there is a need to add some questions or hyoptheses which were tested by perfoming this scientific approach. It is a pity that there is relatively little Connection between the very detailed physical Analysis and the potential consequences

for the biology, e.g. in the introduction the authors mention that These physical processes might influence the species composition of the Phytoplankton. In reality there is just Chlorophyll and one Bloom forming species is mentioned. If there is more Information on the species composition under changing physical conditions of up- and downwelling or intrudions of other water bodies, then this would Support the Quality of the paper. A final Point is the Quality of the Chlorophyll calibrations: different sensors or fluorimeters were used, how good were the intercalibrations between these different measuremnt devices and how stable were they. This would be important Information for other groups dealing with this problem of data conversion. Were the 11 water samples taken over the week or during one transect? In Fig. 6 and 8 legends regarding the o- and x- should be added, to avoid any misunderstanding. All figures are of good Quality and their legends are clear. I did not check the references but at least they are up-to-date. F. Colijn, March 2, 2016
* * *

---

## Referee Comment (RC2) · Dr Svendsen (Referee) · 4 Mar 2016

Comments:

1. Scientific significance:

The paper touch upon a very important issue, namely the effect sub-mesoscale processes have on vertical mixing and thereby supply of nutrients to the euphotic layer impacting primary production and thereby the whole ecosystem. Most numerical models do not have sufficient spatial resolution to handle these processes, and it is a big challenge to find a good way to parameterize these processes in larger scale models. Assessment of the importance of sub-mesoscale processes is not new, however

the compilation of different in situ data to give a 3(4)D view of the processes is to my knowledge quite unique.

2. Scientific quality:

The scientific approach and applied methods are valid. However, my main concern about the paper is how the data are discussed and analysed, the readability. Many time series (wind, hydrography and Chla from different sources) are discussed separately by describing many individual events, and the reader is "drowning" in many event descriptions, having a hard time to connect the links between wind events, up-down-welling, sub-mesoscale features and Chla/prim.prod.

One of the key findings is the typical -2 slope in the horizontal wave number spectra, however it is only in the final discussion they describe what this physically/practically means, namely that sub-mesoscale processes are more energetic than suggested by the quasi-geostrophic theory of turbulence in the ocean interior (maybe obvious to specialists in turbulence). In this respect I would also like to see some quantitative "thoughts" on how much it changes the actual vertical mixing/vertical transports and how we maybe can use this to improve the parameterization in numerical models.

Several places there the direct effect of wind mixing is mentioned. This is more related to the cubed wind speed (than the wind speed), and I suggest including simple time series of cubed wind speed (based on the highest possible resolved data and thereafter averaged to a suitable (daily?) time-resolution).

3. Presentation quality and specific comments

As mentioned above, I would suggest to delete the detailed and lengthy descriptions of the individual data series, and rather focus on fully descriptions of the individual events. This could mean rearranging some of the figures. Some specific comments:

a. Most figures have too tiny text on the axis. Especially I had a very hard time with this on the important Fig.3. This must be changed.

[Figure]

b. Suggest to change "sub-mesoscale" to "sub-mesoscale features" in the title.

c. When first mentioning the spectral slope (-2 versus -3) in the introduction they should say what this actually/physically means (see comment above on this).

d. Suggest adding some names on countries and the Baltic (No American would know where this is on earth).

e. The data from the ferry is assumed to have 160m resolution. I guess this is assuming the ferry always have the same constant speed, independent of weather etc. If this is not the case, how would it affect the results?

f. Related to Figure 2 I would also would like to see time series of cubed wind speed and take this into the discussions/descriptions

g. Fig. 3: In addition to not being able to read the axis text and numbers, it took me a long time to understand the figure. I think some better description on how the data are combined in the different cubes would help.

h. It is mentioned that the spectral slope are up to -3.7, but it is unclear where this is found. (Max values in the table is -2.6)

i. In Fig 6 it is not described which line is what

j. In 3.5 you should also mention that convergence/divergence may rapidly change the concentration of Chla

---

## Author Response (AR1)

Multi-sensor in situ observations to resolve the sub-mesoscale in the stratified Gulf of Finland, Baltic Sea

U. Lips et al.

**Response to Referee 1 (F. Colijn)**

This MS is a very useful contribution to the understanding of physical processes in a coastal sea. The main issue is the use of different observational techniques like moorings, surveys and Ferryboxes to obtain high resolution high frequency data on physical and biological parameters. The paper is well written and easily understandable. There are some language issues which need to be solved by a native speaker and one point of criticism should be taken on board by the authors.

**Response:** English is revised.

The paper is very descriptive, thus there is a need to add some questions or hypotheses which were tested by performing this scientific approach.

**Response:** Descriptive sections are shortened in the revised manuscript. Scientific questions and hypothesis are formulated in a more straightforward way in the revised Introduction section. For instance, the following sentences are added to relevant part of the text: "This suggestion of higher sub-mesoscale activity associated with some types or phases of coastal upwelling has to be analyzed further, and such analysis based on combined Ferrybox, buoy profiler, and Scanfish data is one of the tasks in the present paper." "The hypothesis that under certain mesoscale conditions, such as development and relaxation of coastal upwelling events in a stratified estuary, the sub-mesoscale processes are more energetic than predicted by the theory of quasi-geostrophic turbulence in the ocean interior is tested."

It is a pity that there is relatively little connection between the very detailed physical analysis and the potential consequences for the biology, e.g. in the introduction the authors mention that these physical processes might influence the species composition of the phytoplankton. In reality there is just chlorophyll and one bloom forming species is mentioned. If there is more information on the species composition under changing physical conditions of up- and downwelling or intrusions of other water bodies, then this would support the quality of the paper.

**Response:** We think that a more detailed analysis of impact of physical processes on phytoplankton species composition should be presented in separate papers (as it was done using the data from summer 2010 by Lips and Lips, 2014, referred here). In this paper we mainly have discussed the Chl *a* dynamics (and a vertically migrating species *Heterocapsa triquetra*) when describing the impact of physical processes on phytoplankton.

A final point is the quality of the chlorophyll calibrations: different sensors or fluorimeters were used, how good were the intercalibrations between these different measurement devices and how stable were they. This would be important information for other groups dealing with
this problem of data conversion. Were the 11 water samples taken over the week or during
one transect?
**Response:** Since the chlorophyll *a* fluorescence readings are, besides the chlorophyll *a*
content, influenced by many other factors we have implemented the routines to calibrate the
sensors by regular laboratory analysis of water samples. For the Ferrybox system, the samples
are collected along the ferry route once a week. During different years, depending on the aims
of the measurements, from 11 to 17 samples are collected weekly (once a week). We tried to
be more precise in the revised manuscript when describing this procedure. For the buoy
profiler, the sampling for sensor calibration is conducted bi-weekly and for the Scanfish it is
done in association to each survey. We have used three different sensors (SCUFA, Turner
Design; Seapoint fluorimeter; and TriOS microFlu-chl-A fluorimeter) in the present study.
Seapoint and Trios sensors have been quite stable over the years and for summer conditions
(characterized by certain phytoplankton species composition) only one conversion equation
was used. Seasonally fixed conversion equations were used for the Ferrybox fluorescence
sensor (as described in the manuscript). We consider that the data acquired with the three
sensors fit quite well with each other as seen, for instance, in Fig. 10. A more thorough
analysis of performance of chlorophyll sensors attached to the autonomous systems is a topic
in a separate study (the results will be available soon).
In Fig. 6 and 8 legends regarding the o- and x- should be added, to avoid any
misunderstanding.
**Response:** Explanations added to the figure legends.
All figures are of good quality and their legends are clear. I did not check the references but at
least they are up-to-date.

Ocean Sci. Discuss., doi:10.5194/os-2016-5-RC1, 2016

Multi-sensor in situ observations to resolve the sub-mesoscale in the stratified Gulf of
Finland, Baltic Sea
U. Lips et al.
**Response to Referee 2 (E. Svendsen)**

1. Scientific significance:
The paper touch upon a very important issue, namely the effect sub-mesoscale processes have
on vertical mixing and thereby supply of nutrients to the euphotic layer impacting primary
production and thereby the whole ecosystem. Most numerical models do not have sufficient
spatial resolution to handle these processes, and it is a big challenge to find a good way to
parameterize these processes in larger scale models. Assessment of the importance of sub-
mesoscale processes is not new, however the compilation of different in situ data to give a
3(4)D view of the processes is to my knowledge quite unique.

2. Scientific quality:
The scientific approach and applied methods are valid. However, my main concern about the
paper is how the data are discussed and analysed, the readability. Many time series (wind,
hydrography and Chla from different sources) are discussed separately by describing many
individual events, and the reader is "drowning" in many event descriptions, having a hard
time to connect the links between wind events, up-downwelling, sub-mesoscale features and
Chla/prim.prod.

**Response:** We agree that the description of time series was too lengthy, and the text is
shortened in the revised manuscript. We tried to present the data in a more readable form, e.g.
to have links between the different sub-chapters and figures. See also the response below
where it is justified why we prefer to keep the original structure of the manuscript.

One of the key findings is the typical -2 slope in the horizontal wave number spectra, however
it is only in the final discussion they describe what this physically/practically means, namely
that sub-mesoscale processes are more energetic than suggested by the quasi-geostrophic
theory of turbulence in the ocean interior (maybe obvious to specialists in turbulence). In this
respect I would also like to see some quantitative "thoughts" on how much it changes the
actual vertical mixing/vertical transports and how we maybe can use this to improve the
parameterization in numerical models.

**Response:** This is a very relevant comment. Nevertheless, we think that to have reliable
estimates of changes in vertical mixing/transport and to propose improved parameterization in
numerical models is too large topic to be included in the present paper. We have plans to
conduct such analysis and to present the result in a separate paper.

Several places there the direct effect of wind mixing is mentioned. This is more related to the
cubed wind speed (than the wind speed), and I suggest including simple time series of cubed
wind speed (based on the highest possible resolved data and thereafter averaged to a suitable
(daily?) time-resolution).

**Response:** Times series of wind vectors is replaced by time series of wind stress vectors. See
the comment below. Time resolution of the used data series is 3 hours.

3. Presentation quality and specific comments
As mentioned above, I would suggest to delete the detailed and lengthy descriptions of the
individual data series, and rather focus on fully descriptions of the individual events. This
could mean rearranging some of the figures.

**Response:** We prefer to keep the structure of the paper as it was in the submitted version.
Nevertheless, we agree that the description of time series was too lengthy, and the text is
shortened in the revised manuscript. The decision to keep the original structure is justified by
the main aim of the paper to reveal general statistical characteristics of sub-mesoscale
features/variability and relate them to the mesoscale background. The individual events are
analyzed in separate papers for some data (e.g. summer 2010 results are presented by Lips and
Lips, 2014) or will be a subject of next papers (e.g. events in summer 2012).

Some specific comments:
a. Most figures have too tiny text on the axis. Especially I had a very hard time with this on
the important Fig.3. This must be changed.

**Response:** Fig. 3 is revised.

b. Suggest to change "sub-mesoscale" to "sub-mesoscale features" in the title.

**Response:** Done.

c. When first mentioning the spectral slope (-2 versus -3) in the introduction they should say
what this actually/physically means (see comment above on this).

**Response:** Done. The sentence introducing this issue is complemented in the revised
manuscript. It reads: "It has been shown that if the spatial resolution of numerical models was
increased the spectral slope converted rather to -2 than -3 (Capet et al., 2008) suggesting that
sub-mesoscale processes play an important role in the energy cascade from larger to smaller
scales."

d. Suggest adding some names on countries and the Baltic (No American would know
where this is on earth).

**Response:** Done.

e. The data from the ferry is assumed to have 160m resolution. I guess this is assuming the
ferry always have the same constant speed, independent of weather etc. If this is not the case,
how would it affect the results?

**Response:** The ferry speed certainly influences the data quality and calculation results. For
instance, the changes in the speed cause changes also in the flow through time of water
through the sea chest and, thus, the time lag between the water intake and actual
measurements (as described in the manuscript). One of the advantages of using data from a
regular ferry line is that they always try to keep the schedule, which means also the speed
along the ferry route. In a few occasions, the ferry speed was clearly higher than an average of

15-16 knots. Since the system also records the ferry speed as a background parameter, we
were able to identify those occasions and did not include such data in the analysis.
f. Related to Figure 2 I would also would like to see time series of cubed wind speed and take
this into the discussions/descriptions
**Response:** Times series of wind vectors is replaced by time series of wind stress vectors. We
prefer to present wind stress instead of cubed wind speed since wind stress is the main forcing
behind kinetic energy in the sea. The aim is to show that the sub-mesoscale processes play an
important role in the energy cascade from larger to smaller spatial scales.
g. Fig. 3: In addition to not being able to read the axis text and numbers, it took me a long
time to understand the figure. I think some better description on how the data are combined in
the different cubes would help.
**Response:** Done.
h. It is mentioned that the spectral slope are up to -3.7, but it is unclear where this is found.
(Max values in the table is -2.6)
**Response:** The slopes estimated based on single crossings reached the value -3.7 (see Fig. 4).
In the table average values for certain periods are given.
i. In Fig 6 it is not described which line is what
**Response:** Explanation is added in the figure legend.
j. In 3.5 you should also mention that convergence/divergence may rapidly change the
concentration of Chla
**Response:** Relevant text about the role of the convergence of surface waters (Chl a) and
possible impact of re-stratification is given in the Discussion section.

[revised manuscript text omitted]

regression line equation: Chl *a* = 1.06 x F – 4.11 ($r^2$ = 0.80, p < 0.05). Data only from evening
crossings were used to diminish the fluorescence quenching effect.
**3. RESULTS**
**3.1 Forcing and general features**
The study period in July-August of 2009-2012 was characterizsed by distinct inter-annual
differences in wind conditions and distribution patterns of temperature and salinity in the
central part of the Gulf of Finland. Based on HIRLAM wind data, the average wind speed in
July-August 2009-2012 in the GoFulf of Finland area was 6.0 m s$^{-1}$. and the prevailing wind
direction was from the south-southeast with an average velocity of the airflow of 1.4 m s$^{-1}$.
While the winds from the southwest prevailed in July-August 2009 and 2012 (average
direction from 217° and 214°, respectively), the dominating wind direction was from the
southeast in 2010 and 2011 (average direction from 160° and 122°, respectively). Both in
2010 and in 2011 the monthly average wind direction differed between the two analyzsed
months being from 192° in July and from 115° in August 2010 and from 73° in July and from
177° in August 2011.
On the synoptic scale (several days – a couple of weeks), mostly westerly wind pulses
occurred in 2009 except a period in the first half of July with a relatively strong wind pulse
from the south-southeast (see time series of wind stress vectors in Fig. 2). In 2010, moderate
winds from the southwest were prevailing in the first half of July while several wind pulses
from the east, northeast and south occurred during the rest of the study period. Typical for
2011 was a consecutive appearance of relatively strong wind pulses from southwest and from
east-and northeast. In 2012, westerly winds clearly prevailed with only two short periods when the wind pulses from the east (early July) and northeast (second half of August)

occurred.

Figure 2. Temporal changes in wind stress during the study period of 29 June – 31 August in 2009-2012 based on 3 h average wind measured at the Kalbadagrund meteorological station (Finnish Meteorological Institute) and shown as series of wind stress vectors with a time step of 6 h smoothed using 24-h moving average.

Based on the combined figures of horizontal and vertical distributions of temperature and salinity in July-August 2009 (Fig. 3a and 3b), the following characteristic features could be identified. In the first half of July, an upwelling event developed near the southern coast resulting in large variations of temperature (7.8–17.8 °C) and salinity (4.2–6.1 g kg⁻¹) across
the gulf. Deepening of the thermocline occurred after the upwelling relaxation in the southern
part of the gulf. Shallow and warm upper layer (temperature between 17.3 and 20.4 °C) with
very low variations of salinity across the gulf (between 4.6 and 5.0 g kg⁻¹) appeared in the
study area due to a period of weak winds in the first half of August. Upwelling near the
southern coast occurred in the second half of August with increased across-gulf variability of
temperature (from 12.9 to 17.7 °C) and salinity (4.7–5.6 g kg⁻¹).

[Figure]

Figure 3. Temporal changes in horizontal and vertical distributions of temperature (°C) and salinity (g kg $^{-1}$) in the Gulf of Finland measured by the Ferrybox system between Tallinn and Helsinki and the autonomous buoy profiler at station AP5 from 29 June to 31 August in 2009 (a and b, respectively), 2010 (c and d), 2011 (e and f), and 2012 (g and h). The Ferrybox data are split into two parts at the position of the buoy profiler AP5. The x-axis shows the distance along the ferry route from a starting point off Tallinn harbor at the latitude of 59.48 N.

At the beginning of the study period in 2010, when mainly weak or variable moderate winds prevailed, the variations of temperature (mostly being between 20 and 22 °C) and salinity in the surface layer were very low across the gulf (Fig. 3c and 3d). This calm period was followed by a relatively weak upwelling event off the northern coast and deepening of the thermocline from 10 m to 15 m in the southern part. A strong upwelling event near the southern coast with the high spatial variability of temperature (varying between 11.1 and 21.6 °C) and salinity (varying between 4.0 and 6.3 g kg$^{-1}$) across the gulf occurred in late July. The seasonal thermocline had a much shallower position in 2010 (for the period with available data until early August) than in 2009.

The first half of July 2012 was characterized by relatively low spatial variability of temperature and salinity in the surface layer of the study area (Figs. 3g and 3h). An upwelling event occurred near the northern coast in at the end of July, creating a temperature difference across the gulf from 10.3 to 17.2 °C, and accompanied with the deepening of the seasonal thermocline in the southern part (to 45 meters). After a short period with low variability, the second upwelling event appeared near the northern coast while the surface layer temperature stayed quite high in the rest of the study transect (up to 20 °C). In the period between the two upwelling events, strong intrusions of more saline waters were observed in the subsurface layer at the buoy station. The position of the seasonal thermocline in the southern part of the gulf was the deepest in 2012 among the analyzed years.

**3.2 Lateral variability of temperature in the surface layer**

Overall horizontal variability of temperature characterized as the standard deviation of temperature along the ferry route was varying in quite large ranges in time – from 0.2 °C to 3.7 °C (Fig. 4). High values of standard deviation of temperature in the surface layer were related to the observed coastal upwelling events and, as a rule, the upwelling events near the southern coast resulted in larger spatial variations of temperature than those near the northern coast. During the upwelling event in August 2010 the standard deviation of temperature was as high as 3.7 °C while during the other upwelling events within the study period in July-August 2009-2012, the values of standard deviation of temperature did not exceed 2.5 °C.

Despite  the high temporal variability of standard deviations of temperature calculated on data from single crossings, the average values of standard deviations for the studied four years did not differ much – minimum of 0.71 °C was found in 2009 and maximum of 0.83 °C

in 2010 (Table 1).

[Figure]

Figure 4. Statistical characteristics of the temperature variability in the surface layer of the Gulf of Finland along
the ferry route Tallinn-Helsinki from 29 June to 31 August in 2009, 2010, 2011 and 2012. Standard deviations of
temperature are shown as solid lines and spectral slopes of temperature variance between the horizontal scales of
10 and 0.5 km as dotted lines. The vertical dashed lines denote the borders between the selected characteristic
periods with similar variability patterns (numbers of periods are shown in the upper part of the panels).

[Figure]

Figure 5. Horizontal wavenumber spectra of temperature variance in the surface layer of the Gulf of Finland
calculated using Ferrybox data from the Tallinn-Helsinki ferry line in summers 2009-2012. The bold lines show
the average spectral curve for the entire study period from 29 June to 31 August in each year, and the thin lines
represent the average spectral curves in the selected periods. The numbers of the periods, corresponding to those
marked in Fig. 4 and listed in Table 1, are shown close to each respective spectral curve. The dashed lines
correspond to -2 and -3 slopes.

The calculated horizontal wavenumber spectra of temperature variance had also relatively large variability if to compare the spectra estimated based on data from single crossings. The spectral slope between the lateral scales of 10 and 0.5 km varied between -1.8 and -3.7 (in logarithmic scales). Note that the spectral curves were approximately linear (Fig. 5) between the scales of 15-20 km (the latter corresponds to horizontal wavenumber of 0.05 km$^{-1}$ or in logarithmic scale to -1.3 in Fig. 5) and 0.5 km (corresponds to wavenumber of 2 km$^{-1}$ or in logarithmic scale 0.3 in Fig. 5) – thus, linear approximation of their slopes is feasible. Within the periods of the high spatial variability of temperature, mostly related to upwelling events affecting the distribution of temperature in the surface layer of the Gulf of Finland, the estimated slopes were between -1.8 and -2. When the spatial variability of temperature was low in the surface layer, the slopes varied mostly between -2 and -3 (Fig. 4). At the same time, the average spectra for the entire period under consideration in the studied years were quite close to each other (Fig. 5, bold lines) and the spectral slopes on average were close to -

2 (from -2.1 to -2.2; see also Table 1).

Table 1. Standard deviations of temperature and slopes of wavenumber spectra of temperature variance based on the data collected in the surface layer along the ferry route between Tallinn and Helsinki. Average values for each year over the study period from 29 June to 31 August (to 22 August in 2012) and within the selected periods with similar spatial variability are given. Numbers of the periods correspond to the periods marked in Fig. 4.

| Year No | Dates | Standard deviation (°C) | Spectral slope (10 km – 0.5 km) |
|---|---|---|---|
| **2009** | **29 June – 31 August** | **0.71** | **-2.1** |
| 1 | 29 June – 15 July | 1.26 | -1.9 |
| 2 | 16 July – 14 August | 0.37 | -2.3 |
| 3 | 15 August – 31 August | 0.78 | -1.9 |
| | | | |
| **2010** | **29 June – 31 August** | **0.83** | **-2.2** |
| 1 | 29 June – 18 July | 0.52 | -2.3 |
| 2 | 19 July – 31 July | 1.46 | -2.0 |
| 3 | 1 August – 16 August | 0.48 | -2.2 |
| 4 | 17 August – 24 August | 1.89 | -1.9 |
| | | | |
| **2011** | **29 June – 31 August** | **0.73** | **-2.2** |
| 1 | 29 June – 12 July | 0.93 | -2.1 |
| 2 | 13 July – 25 July | 0.38 | -2.6 |
| 3 | 26 July – 9 August | 1.43 | -1.9 |
| 4 | 10 August – 31 August | 0.34 | -2.2 |
| | | | |
| **2012** | **29 June – 22 August** | **0.76** | **-2.2** |
| 1 | 29 June – 16 July | 0.32 | -2.6 |
| 2 | 17 July – 13 August | 1.16 | -2.0 |
| 3 | 14 August – 22 August | 0.35 | -2.4 |

Based on the presented lateral variability of temperature, some distinct periods when the standard deviation of temperature was high and spectral slope was close to -2 can be distinguished. We selected 3-4 periods with the almost quasi-stationary character of variability in each year to describe quantitatively the character of variability within these periods; the periods are marked in the Fig. 4 by dashed lines.

In 2009, two periods of high spatial variability caused by coastal upwelling events existed: (period (1) in the first half of July when upwelling occurred near the southern coast and period (3) marked in the second half of August when upwelling developed near the northern coast. In Fig. 4). During both periods, the spectral lines had a higher position, and their slopes were shallower than the average for the entire study period in 2009 (see Fig. 5 and Table 1).

In 2010, two periods, which were also associated with thean upwelling events near the northern coast in mid July was followed almost immediately by an event near the southern coast – we kept those events within one(-period (2) and . Later on, a very intense upwelling event occurred near the southern coast – period (4); see Fig. 4), . Both mentioned periods had much higher spatial variability, and the spectral slopes  were shallower than the average in 2010.  All time intervals comprising upwelling events in 2011 (periods (1) and (3); see Fig. 4) and 2012 (period (2); see Fig. 4) were also characterized by a higher position and shallower slope of spectral lines than the lines representing the average for July-August 2011 and 2012. The noticed divergence of spectral slopes from the high-variability and low-variability periods resulted in a clearly larger separation between the spectral curves at the sub-mesoscale than at the mesoscale. While the spectral density of spatial variations of temperature at the spatial scale of 1 km varied more than 1.5 magnitudes, it varied in ranges of one magnitude at the spatial scale of 10 km (Fig. 5).

**3.3 Temporal variability of the vertical stratification**

The vertical distributions of temperature and salinity at the buoy station varied considerably in time similarly to the horizontal distributions of temperature and salinity along the ferry route. The variations were  revealed as changes in the magnitude of vertical gradients, depth of the upper mixed layer and seasonal thermocline, fast deepening or surfacing of the thermocline,  and occurrence of intrusions leading in certain cases to local inversions in vertical salinity distribution (Fig. 3).

Temporal changes in vertical stratification in the Gulf of Finland could be related to the differences in the heat flux through the sea surface and to the prevailing wind forcing that influences both the estuarine circulation alterations and the intensity of vertical mixing (see e.g. Liblik and Lips, 2012). Note that the autonomous buoy station in the present study was located in the southern part of the open Gulf of Finland. Thus, in addition to the seasonal course of stratification and its dependence on the estuarine circulation, the vertical stratification at this location could be significantly influenced both by the upwelling and by the downwelling along the southern coast.

depth increased in July 2010 and 2011 in accordance with the strengthening of the seasonal thermocline (Fig. 6). In July-August of these years, the winds from the southeast prevailed supporting the estuarine circulation and, in turn, keeping up the strong vertical stratification – the maximum of $P = 370$ J m$^{-3}$ was observed  at the beginning of August 2010. This continuous increase of *P* in both years was disrupted only due to the coastal upwelling events (in 2010 also due to a weak downwelling event) leading to rapid changes  in the stratification parameter mostly because of vertical movements of the thermocline. In contrast to 2010 and 2011, the stratification parameter did not increase much during the study window in 2009 and 2012 in accordance with the prevailing southwesterly winds. In 2009, the stratification parameter being relatively high  due to the vertical salinity stratification had the maximum in the first half of August ($P = 300$ J m$^{-3}$) and  decreased rapidly afterwards when the downwelling influence reached the buoy station. In 2012, the vertical stratification in the water layer from the surface to 40 m depth almost vanished at the measurement site

AP5 by 20 July due to a very strong downwelling event, which appeared along the southern coast of the Gulf of Finland. Later on, the stratification at the buoy station strengthened, but the stratification parameter was clearly the lowest in 2012 if compared to the other years due to the deepest position of the seasonal thermocline .

[Figure]

Figure 6. Daily average stratification parameter (solid line with open circles) and intrusion index (dashed line
with crests) estimated for the water column from the sea surface to 40 m depth at the buoy station AP5 in the
central Gulf of Finland from 29 June to 31 August in 2009-2012. Location of the buoy station is shown in Fig. 1.

Vertical profiles of temperature and salinity collected at the buoy station often exposed variability with vertical scales of a few to ten meters that could be interpreted as intrusions related to the sub-mesoscale dynamics. Since the temperature was the main contributor to the vertical density distribution in the seasonal thermocline, such intrusions could create local inversions in the vertical distribution of salinity as mentioned above and seen in Fig. 3. The calculated intrusion index showing how much the vertical stratification is weakened due to local salinity inversions varied mostly between 0 and 0.05.  However, every year one or a few periods were detected when the index exceeded 0.05, whereas the maximum index value obtained on 1-2 August 2012 reached 0.36.

In 2009, the only period with relatively high intrusion index values was detected during and just after the period of estuarine circulation reversal (Liblik and Lips, 2012).

The maximum of the intrusion index coincided with the last day of the upwelling event near the northern coast that was followed by the event near the southern coast and rapid decrease of the intrusion index. In 2011, the index values > 0.05

were detected a few times in July,  whereas the highest values on 12-14 August (exceeding

0.24 on 14 August) were related to the relaxation of an intense upwelling event near the southern coast and short-term deepening of the thermocline at the buoy station during a weak upwelling event near the northern coast. The mentioned highest intrusion index value on 1-2

August 2012 was detected within the period when two consecutive major upwelling events occurred near the northern coast, whereas this maximum emerged between the upwelling events just before the second one.

Thus, the intrusions were most intense (in the sense of salinity inversions) at the buoy station AP5 in connection to the relaxation of upwelling events near the southern coast, development of upwelling events near the northern coast and estuarine circulation reversals.

All these situations correspond to the periods when the thermocline was deepening or was already at a deep position at the buoy station in the southern part of the open Gulf of Finland.

In addition, the stratification parameter values were low or decreasing when the temporal maximum of intrusion index was detected. When relating intrusion index values with the lateral variability in the surface layer then one could conclude that the found temporal maxima of intrusion index corresponded to the periods of moderate lateral variability in the surface layer. Nevertheless, during such periods, the slopes of horizontal wavenumber spectra of temperature variance were close to -2 as  during the periods of high lateral variability and approximately a week before the highest intrusion index values, the lateral variability in the surface layer was also high (Figs. 4 and 6).

**3.4 Spatial variability in the thermocline**

We analyzed the data of Scanfish surveys in the Gulf of Finland conducted across the gulf in the open, deeper part and along the  southern coast in summers 2010

and 2012. The hydrographic background of surveys in 2010 is characterized by the
development of a weak upwelling along the northern coast of the gulf on 22 July 2010, a
strong upwelling event along the southern coast on 27 July 2010 and relaxation of it by 2
August 2010 when the last survey was conducted (see Fig. 3c,d). In summer 2012, when the
upwelling events along the northern coast dominated, the survey on 4 July 2012 characterizes
the situation before those upwelling events, on 20 July 2012 the development of upwelling
and on 31 July 2012, which was conducted along the gulf, the situation related to a temporal
relaxation of upwelling (see Fig. 3g,h).

[Figure]

[Figure]

Figure 7. Vertical sections of temperature, salinity and density anomaly measured using the Scanfish on 22 July
2010 (a), 27 July 2010 (b), 2 August 2010 (c), 4 July 2012 (d), 20 July 2012 (e), and 31 July 2012 (f). The
corresponding Scanfish tracks are shown in Fig. 1.

A clear cross-gulf inclination of the thermocline was revealed on 22 July 2010 (Fig. 7a), although the Scanfish section did not reach the upwelling area near the northern coast. At the same time, the isopycnals had opposite inclination below the 20 m depth resulting in a weakening of the vertical stratification from south to north (Fig. 8a). A

well-pronounced, less saline water zone with a width of less than 5 km was observed in the surface layer. The extension of an associated intrusion of lower salinity in the thermocline was wider in the horizontal dimension and its thickness, decreasing from north to south in accordance with the strength of the vertical stratification, was less than 5 m.

[Figure]

Figure 8. Stratification parameter (open circles) and intrusion index (crests) estimated for the water column from the sea surface to 40 m depth along the Scanfish tracks on 22 July 2010 (a), 27 July 2010 (b), 2 August 2010 (c), 4 July 2012 (d), 20 July 2012 (e), and 31 July 2012 (f). The corresponding Scanfish tracks are shown in Fig. 1.

 The along-gulf Scanfish section on 27 July 2010 from the buoy station AP5 to the south‑west crossed the meandering upwelling front (Fig. 7b). The observed variability is characterized by clear mesoscale meanders of the front with spatial scales of 10-15 km, strong stratification at the warm, less saline side of the front and much weaker stratification at the cold, more saline side of it, and very low intrusion index almost along the entire section (Fig. 8b). A remarkable variability at the sub-mesoscale, also resulting in intrusions of waters with different salinity seen in Fig. 7c and expressed in high values of intrusion index (Fig. 8c), was observed on 2 August 2010. A less saline water zone was well visible almost at the same location as on 22 July 2010, but the surface layer salinity in it was
much lower – less than 4.5 g kg$^{-1}$ (Fig. 7c).

upper 40 m water layer were clearly lower in summer 2012 (Fig. 7d-f) than in summer 2010
(Fig. 7a-c). A less saline water zone in the central part and slightly stronger vertical
stratification in the southern part of the section were observed on 4 July 2012 (Fig. 7d and
8d). Development of the upwelling along the northern coast and downwelling in the southern
part caused strong inclination of the thermocline across the gulf and a clear strengthening of
vertical stratification from south to north on 20 July 2012 (Fig. 7e and 8e). In the area of
strong inclination of the thermocline, relatively large horizontal gradients of salinity and
intense sub-mesoscale variability, also seen  as intrusions of waters with different salinity
(Fig. 7e), were observed. On 31 July 2012, the Scanfish survey revealed a mesoscale eddy
like feature (Fig. 7f), which could be formed in the process of downwelling relaxation as also
observed  earlier along the southern coast of the gulf (in its mouth area; see Lips et al.,
2005). This mesoscale feature was characterized  by relatively weak vertical stratification
in its central part and high intrusion index values, especially at its periphery (Fig. 8f). Note
that the pronounced intrusion of more saline waters detected at the western end of the section
was also registered at the buoy station during several days (Fig. 3h).

[Figure]

Figure 9. Horizontal wavenumber spectra of temperature variance in the sub-surface layer of the Gulf of Finland
calculated using Scanfish data from 22 July 2010, 27 July 2010, 2 August 2010, 4 July 2012, 20 July 2012, and
July 2012. The bold lines show the average spectral curve for each survey and the thin lines represent the
spectral curves in the selected layers with the thickness of 10 m. The central depth values of the selected layers
are indicated at the left side of panels and the estimated spectral slopes for the average spectral curve at the right
of panels. The dashed lines correspond to -5/3 and -2 slopes.

The spectral slopes between the horizontal scales of 10 and 1 km for spectra averaged over depth intervals with the thickness of 10 m were mostly shallower than -2; the average values of spectral slopes — the numbers corresponding to the mean curves for each survey (shown in

Fig. 9) varied between -1.7 and -2.0. The cases with the lowest mesoscale variability had also the shallowest spectral slopes. The slopes close to -2 were obtained for the surveys with the most pronounced mesoscale features. Local vanishing of the spectral slope could be detected between the horizontal scales from 3 to 1 km on 2 August 2010 and at scales from 3 to 2 km on 31 July 2012. High intrusion index values were characteristic for the both mentioned surveys, especially for the survey on 31 July 2012 (Fig. 8c and f), which is in accordance with the index estimates based on the buoy profiler data (Fig. 6d). High intrusion index values were also found  for the survey on 20 July 2012 when a patch of more saline waters appeared in the sub-surface layer at the warm side of the upwelling front (Fig. 7d and 8d).

**3.5 Consequences to chlorophyll *a* dynamics**

Temporal variability of chlorophyll *a* at the scales of days is usually much higher than that of temperature and salinity since in addition to the advection and mixing, the phytoplankton growth (and decay) could increase (decrease) the biomass and consequently chlorophyll *a* content rapidly. Despite  such high variability and other factors that could influence the comparability of acquired chlorophyll *a* fluorescence data, e.g. fluorescence quenching, the presented combined plots of changes in horizontal and vertical distributions agree reasonably well (Fig. 10).

[Figure]

Figure 10. Temporal changes in horizontal and vertical distribution of chlorophyll *a* (mg m$^{-3}$) in the Gulf of Finland measured by the Ferrybox system between Tallinn and Helsinki and the autonomous buoy profiler at station AP5 from 29 June to 31 August in 2010 (a) and 2012 (b). The Ferrybox route and the location of station AP5 are shown in Fig. 1.

In the first half of July 2010, a sub-surface bloom developed in the southern part of the Gulf of Finland, which occasionally was also seen  in the surface layer with higher chlorophyll *a* values off the southern coast (Fig. 10a). When an upwelling event along the southern coast started to dominate (see Fig. 3c), the chlorophyll *a* content decreased in the southern part and increased in the northern part of the study area. Before the bloom near the northern coast, relatively deep chlorophyll *a* maxima were detected at the buoy station at the depths below 20 m (since the maxima layers were thin they are not well seen in Fig. 10) and at the Scanfish section on 22 July 2010, especially at its northern part (Fig. 11). When the upwelling developed along the southern coast of the gulf, the sub-surface chlorophyll *a* maxima were situated at a shallower position in the warmer side of the front, but they almost disappeared by 2 August 2010 after relaxation of the upwelling event. In-between, the bloom
developed near the northern coast in the convergence/downwelling area. It was suggested that
this bloom could be related to the observed sub-surface maxima of chlorophyll *a*, which
contained the dinoflagellate *Heterocapsa triquetra* in very high abundances (Lips and Lips,
2014).

[Figure]

Figure 11. Vertical sections of chlorophyll *a* content measured using the Scanfish on 22 July 2010, 27 July 2010,
August 2010, 4 July 2012, 20 July 2012, and 31 July 2012. The corresponding Scanfish tracks are shown in
Fig. 1.

In 2012, the chlorophyll *a* content was higher in the northern half of the study area than in
its southern part in the first half of July. Later, when the two consecutive upwelling events
appeared along the northern coast, the highest chlorophyll *a* values were observed close to the
southern coast. At the buoy station, where the downwelling influence was visible (see Fig.
3g), the chlorophyll *a* content also increased  in the sub-surface layer. Occasionally, high
chlorophyll *a* values were detected close to the upwelling front in the northern part of the
study area. While  the sub-surface maxima of chlorophyll *a* were
observed at the buoy station and  the Scanfish transect at the beginning of July 2012 (~~4 July
2012;~~ Fig. 11), they disappeared when the upwelling events occurred near the northern coast.
During the upwelling development, both the chlorophyll *a* content in the surface layer and the
thickness of the surface layer with elevated chlorophyll *a* increased  from north to south.

However, those tendencies had pronounced intermittency at lateral scales of one to a few kilometers. Similar distribution pattern was registered in the phase of the upwelling/downwelling relaxation on 31 July 2012.

**4. DISCUSSION AND CONCLUSIONS**

Many earlier studies have noticed that proper in situ measurements to reveal sub-mesoscale features are difficult to organize since the variability both in space and in time has to be tackled simultaneously (e.g. Hosegood et al., 2008, Niewiadomska et al., 2008, Pietri et al.,

2013). Especially challenging are the investigations of sub-mesoscale processes in the coastal, relatively shallow but vertically stratified sea areas where the characteristic baroclinic

Rossby radius is in order of a few kilometers, as in the Gulf of Finland – 2-5 km (Alenius et al., 2003). We suggest that the most promising approach to solve the problem is to apply a combination of autonomous and research vessel based devices, such as Ferryboxes, moored profilers, underwater autonomous vehicles (gliders) and towed undulating instruments (Scanfish).

Simultaneous temporal changes that could be related to mesoscale processes are clearly seen in horizontal and vertical distributions of temperature and salinity presented in Fig. 3.

The sub-mesoscale features such as upwelling filaments were also registered simultaneously by both systems, for instance in the first half of July 2009 and on 24 July 2010. Furthermore, the buoy profiler and  Scanfish simultaneously detected intrusions of waters with different salinity in the thermocline layer. Thus, the application of high-resolution autonomous and towed devices, which measure horizontal and vertical distributions of environmental parameters, makes it possible to detect meso- and sub-mesoscale features and quantitatively estimate their properties. In the present study, the underwater gliders were not applied, but they have been successfully tested in the Baltic Sea (e.g. Karstensen et al., 2014).

If the high-resolution measurements have a large enough coverage in space and time, one is able to reveal statistical parameters of sub-mesoscale variability. In turn, this would lead to an improved parameterization of sub-grid processes in the numerical models that has been considered as a problem in the modelling of the relatively shallow, but stratified Baltic Sea sub-basins (Tuomi et al., 2012; Omstedt et al., 2014). It also allowed us to display some general features of spatiotemporal variability of temperature and salinity in the study region –

the central Gulf of Finland. The upwelling events along the southern coast were associated with higher horizontal variability of temperature in the surface layer than those along the northern coast (Kikas and Lips, 2015; Liblik and Lips, 2016). In the case of prevailing westerly winds, the seasonal thermocline has a deeper position and the vertical gradient of salinity is weaker than in the case of easterly winds (Liblik and Lips, 2012).

One of the questions addressed in the present study was whether the wavenumber spectra of temperature variance convert to -3 slope predicted by the theory of quasi-geostrophic turbulence in the ocean interior (Charney, 1971) or rather to -5/3 slope predicted by the theory of surface quasi-geostrophic turbulence (Held et al., 1995). We found that the wavenumber spectra of temperature variance in the surface layer had slopes varying mostly between -1.8

and -3.7 estimated for the lateral scales from 10 to 0.5 km. Nevertheless, when high variability at the mesoscale, i.e. pronounced mesoscale features, were observed, the spectral slopes were shallower than -2. Similar tendency towards -2 slope was obtained for the wavenumber spectra of temperature variance in the thermocline layer between the spatial scales of 10 and 1 km. These estimates were very stable over the four years of Ferrybox measurements and all Scanfish surveys analyzed in the present study.

Such conversion of wavenumber spectra of temperature variance to -2 slope has been identified earlier in other sea areas by high-resolution modelling (e.g. Capet et al., 2008) and in situ measurements (e.g. Hodges and Rudnick, 2006). Based on remote sensing altimeter data, it is shown that sea level wavenumber spectra also correspond well to the surface quasi- geostrophic theory (Le Traon et al., 2008). In a recent study, Kolodziejczyk et al. (2015)

showed that if the surface density is analyzed then the -2 spectral slope is obtained in summer conditions when the salinity and temperature variations do not compensate each other (in north-eastern subtropical Atlantic Ocean). We have used temperature data to estimate potential energy wavenumber spectra assuming that mostly temperature determines the density in the upper layer (including the seasonal thermocline) in the Gulf of Finland in summer. It has to be noted that the wavenumber spectra of density variance corresponded to -

2 slope as well when the spatial variability was dominated by coastal upwelling events.

According to these findings, the sub-mesoscale processes have to be more energetic than suggested by the quasi-geostrophic theory of turbulence in the ocean interior. Thus, the observed high lateral variability of temperature in the surface layer and associated -2 spectral slopes suggest a significant role of sub-mesoscale processes in vertical exchanges in the stratified Gulf of Finland and similar sea areas.

The lateral variability of temperature in the sub-surface layer was the highest during the surveys when the upwelling events either off the southern or off the northern coast occurred (Scanfish sections on 27 July 2010 and  20 July 2012). Higher intrusion index values in the sub-surface layer were also found at the Scanfish sections in relation to the development and relaxation of coupled upwelling/downwelling events, except at the section crossing the meandering upwelling front on 27 July 2010. One could suggest that the intrusion index (counted as a sum of salinity inversions) indicates the presence of the layered flow structure and thus, the intensity of lateral mixing. When analyzing the characteristics of coastal upwelling, Kikas and Lips (2015) suggested that two types of upwelling events could be identified. During the event on 18-27 July 2012, no pronounced upwelling front was detected, rather a gradual decrease of the surface temperature from the open sea towards the coast with remarkable variability at the sub-mesoscale was observed. It was suggested that such upwelling events could develop when the wind forcing is weaker than required to generate an

Ekman drift in the entire upper layer and consecutive surfacing of the thermocline.

The observed salinity intrusions at the Scanfish section on 20 July 2012 support the above suggestion by Kikas and Lips (2015). The seasonal thermocline was relatively deep in July

2012 and most probably, the observed salinity intrusions were formed as a response to the winds favorable for the upwelling near the northern coast. Consequently, in such conditions, the lateral mixing is enhanced as the transport of waters with different characteristics upward and downward along the inclined isopycnals. In turn, it could result in enhanced vertical (diapycnal) mixing of waters at laterally distant places from their origin. We suggest that sub- mesoscale dynamics and layered flow structure contribute significantly to the lateral and vertical mixing in the stratified sea areas under variable wind forcing.

The highest values of intrusion index were registered at the buoy station in late July – early

August 2012 and at the Scanfish section on 31 July 2012 during the relaxation of the downwelling near the southern coast. Apart of this major sub-mesoscale structure, similar intrusions visible in the vertical salinity distribution at the buoy station were quite frequent in summers 2009-2012 in the seasonal thermocline layer – e.g. in late July – early August 2009, in mid-August 2011 and in July and August 2012 (Fig. 3). In addition, waters with slightly lower salinity were occasionally seen at the buoy station in July 2010 and clear evidences are provided by the Scanfish surveys on 22 July 2010 and 2 August 2010 that such intrusions of low salinity waters in the upper part of the seasonal thermocline originated from patches of lower surface salinity in the central gulf. At least in two occasions we could detect clear inclination of salinity intrusions in relation to the isopycnals – on 2 August 2010 in the southern part of the section and on 20 July 2012 in the central part of it. This finding is similar to the observations by Pietri et al. (2013) in the upwelling system off southern Peru, where they suggested that observed sub-mesoscale features could be the result of the stirring by the mesoscale circulation. Note that the sub-surface chlorophyll *a* maxima registered  in the
northern part of the Scanfish section on 22 July 2010 were also inclined in relation to the
isopycnals (see Fig. 11).

Two examples of bloom development in the near coastal convergence zone were shown in
the present study – in late July 2010 near the northern coast and in July and August 2012 near
the southern coast (Fig. 10). Lips and Lips (2014) suggested that the bloom near the northern
coast in 2010 could be related to the sub-surface maxima of chlorophyll *a*, which contained
the vertically migrating dinoflagellate *Heterocapsa triquetra* in very high abundances. Similar
development of the biomass peak with a relatively high share of this vertically migrating
species in the surface layer was observed in the same area also in August 2006 (Lips and
Lips, 2010). The highest biomass and chlorophyll *a* content in that convergence zone was
associated with the locally higher location of isopycnals, thus. with the stratified conditions in
the surface layer, although in the downwelling area.

The Scanfish surveys conducted during the downwelling event and its relaxation at the end
of July 2012 did not show high chlorophyll *a* content in the sub-surface layer. However, the
data both from the buoy station and from the Scanfish surveys registered clearly enhanced
chlorophyll *a* content in the surface layer with quite a large intermittency in the chlorophyll *a*
content and layer thickness with enhanced chlorophyll *a* content (Figs. 10 and 11). Note that
the blooms lasted relatively long time (about  ten days). and the highest biomass
(chlorophyll *a* content) was not observed near the mesoscale upwelling front where the largest
vertical velocities could be expected (e.g. Thomas and Lee, 2005). Levy et al. (2012) showed
that the sub-mesoscale processes have large-scale effect on phytoplankton growth in the
ocean, which could be seen at larger scales and distant places. An improvement in the
resolution of ocean circulation models has resulted in more energetic motions not only close
to the large scale (or mesoscale) fronts but rather in the surface layer of the whole modelling
domain (Capet et al., 2008; Levy et al., 2010).

We suggest that the maintenance of the bloom, which could not be explained by pure
convergence due to the Ekman drift in the surface layer, must benefit from other processes
feeding the surface layer with nutrients and/or biomass. The ageostrophic sub-mesoscale
processes could be responsible for re-stratification of the surface layer, vertical transport and
thus, also for growth enhancement (Levy et al., 2012). This conclusion supports the concept
that the vertical exchanges related to the mesoscale processes (eddies) are enhanced due to the
sub-mesoscale activity and not only in the vicinity but also far off the mesoscale features
(Klein and Lapeyre, 2009).

The results of the present study can be concluded as follows. The analysis of high-resolution data from summers 2009-2012 revealed pronounced sub-mesoscale features in the surface and subsurface layer, e.g. upwelling and downwelling filaments and intra-thermocline intrusions with spatial scales of a few kilometrers (typical baroclinic Rossby radius in the Gulf of Finland is 2-5 km). The horizontal wavenumber spectra of temperature variance estimated between the lateral scales of 10 and (1)0.5 km had the slopes close to -2 both in the surface layer and in the seasonal thermocline. It shows that the ageostrophic sub-mesoscale processes contribute considerably to the energy cascade in this stratified sea basin. We showed that the role of sub-mesoscale processes iswas significant especially in the conditions of changing wind forcing, e.g. during the development and relaxation of coastal upwelling and downwelling events. We suggest that the sub-mesoscale processes play a major role in feeding surface blooms in the conditions of coupled coastal upwelling and downwelling events in the Gulf of Finland.

**Acknowledgements**

The work was supported by institutional research funding (IUT19-6) of the Estonian Ministry of Education and Research, by the Estonian Science Foundation grant no. 9023 and by EU Regional Development Foundation, Environmental Conservation and Environmental Technology R&D Programme project VeeOBS (3.2.0802.11-0043).